



# Assessment of altimetry using ground-based GPS data from the 88S Traverse, Antarctica, in support of ICESat-2

Kelly M. Brunt[1,2], Thomas A. Neumann[2], and Christopher F. Larsen[3]

[1]Earth System Science Interdisciplinary Center (ESSIC), University of Maryland, College Park, MD, USA
[2]NASA Goddard Space Flight Center, Greenbelt, MD, USA
[3]Geophysical Institute, University of Alaska, Fairbanks, Fairbanks, AK, USA

Correspondence to: Kelly M. Brunt (kelly.m.brunt@nasa.gov)

**Abstract.** We conducted a 750 km kinematic GPS survey, referred to as the 88S Traverse, based out of South Pole Station, Antarctica between December 2017 and January 2018. This ground-based survey was designed to validate spaceborne
altimetry and airborne altimetry developed at NASA. The 88S Traverse intersects 20% of the ICESat-2 satellite orbits on a route that has been flown by 2 different Operation IceBridge airborne laser altimeters: the Airborne Topographic Mapper (ATM; 26 October 2014) and the University of Alaska, Fairbanks (UAF) Lidar (30 November and 3 December, 2017). Here we present an overview of the ground-based GPS data quality and a quantitative assessment of the airborne laser altimetry over a flat section of the ice-sheet interior. Results indicate that the GPS data are internally consistent ($1.1 \pm 4.1$ cm). Relative
to the ground-based 88S Traverse data, the elevation biases for ATM and the UAF Lidar range from -9.5 to 3.6 cm, while surface measurement precisions are equal to or better than 14.1 cm. These results suggest that the ground-based GPS data and airborne altimetry data are appropriate for the validation of ICESat-2 surface elevation data.

## 1 Introduction

The Ice, Cloud, and land Elevation Satellite-2 (ICESat-2) is a next-generation laser altimeter developed by the National
Aeronautics and Space Administration (NASA) and set to launch in September 2018 (Markus et al., 2017). ICESat-2 will carry a single instrument, the Advanced Topographic Laser Altimeter System (ATLAS), a 6-beam photon-counting system using <2 ns, 532 nm wavelength pulses with a 10 kHz repetition rate. ICESat-2 will continue NASA's multidecade effort to measure changes in the polar regions (Markus et al., 2017; Webb et al., 2012; Zwally et al., 2011), with mission requirements that include the determination of ice-sheet elevation change rates to an accuracy of less than or equal to 0.4 cm a$^{-1}$ (Markus et al.,
2017).

Plans for the post-launch validation of ICESat-2 elevation data products include utilizing both ground-based and airborne elevation datasets. The relatively short ground-based datasets, such as presented here, will provide error assessments for airborne surveys, such that longer airborne surveys can then be designed with sufficient length scales to provide the data volume required for meaningful statistics of satellite data validation. The ground-based activities include the kinematic GPS



validation efforts at Summit Station, Greenland (Brunt et al., 2017) and airborne activities, such as those associated with NASA's Operation IceBridge (OIB; Koenig et al., 2010), which includes a lidar as part of the instrument suite.

In support of the ground-based component of ICESat-2 data validation, we conducted a 750 km traverse based out of South Pole Station (Fig. 1), referred to as the 88S Traverse (28 December 2017 – 10 January 2018). Kinematic GPS data collected along this traverse was used to validate airborne data and will ultimately be used for validation of ICESat-2's space-borne datasets.

ICESat-2 will have 1387 unique orbits over a 91-day orbital cycle (i.e., all 1387 unique tracks are sampled every 91 days, or 4 times per year). The orbit has an inclination of 92º, allowing for data collection between 88º north and south. Since ICESat-2 is a 6 beam instrument, we refer to the imaginary centerline of the beam pattern as the reference ground track for each of the 1387 tracks. The 88S Traverse was designed specifically to include 300 km of data along the 88º S line of latitude, which is the latitude limit of ICEsat-2 and where the ICESat-2 reference ground tracks will converge. This 300 km traverse along 88º S represents 20% of the total length of this line of latitude; the traverse route will therefore intersect 20% (277) of the 1387 ICESat-2 reference ground tracks. Because of Earth rotation, time-sequential ground tracks are not geographically sequentially spaced along the 88º S line of latitude. Therefore, the 20% of the 1387 ICESat-2 unique tracks intersected by this survey are spaced randomly in time over the 91-day orbital cycle. Further, since the ground tracks intersected by the 88S Traverse are spread evenly throughout the 91-day orbital cycle, data from the 88S Traverse mitigates weather limitations (i.e., cloud cover) that have had an impact on other validation campaigns, which utilize only a few tracks within a small area of interest (e.g., Fricker et al., 2005).

The design of the 88S Traverse was based on validation studies associated with ICESat and OIB research. These include studies from the Summit Station Traverse (Brunt et al., 2017; Siegfried et al., 2011) and the Norway-USA East Antarctic Traverse (Kohler et al., 2013). Brunt et al. (2017) used data from an 11 km ground-based kinematic GPS traverse near Summit Station, in the center of the Greenland Ice Sheet, to assess the elevation bias and surface measurement precision of OIB laser altimeters, including the Airborne Topographic Mapper (ATM) and Land, Vegetation, and Ice Sensor (LVIS). Using Precise Point Positioning (PPP) post-processing methods, elevation biases for these altimeters ranged from -10.8 to 8.2 cm, while surface measurement precisions were equal to or better than 8.7 cm. Their results suggest that for a flat, relatively smooth and homogeneous surface, these altimeters provide consistent results, which are required for an airborne component of an ICESat-2 validation strategy. Kohler et al. (2013) collected 5000 km of ground-based kinematic GPS data across East Antarctica, in 2 different vehicles and over the course of 2 different field campaigns, for direct comparison with ICESat elevation data. Using PPP post-processing methods, elevation biases for ICESat ranged from -12 to -2 cm, while surface measurement precisions were equal to or better than 15.8 cm. Their results were based on cross-over analysis between ground-based measurements and the last 2 years of ICESat data.

Here we present results from the first 88S Traverse and show that 1) this part of Antarctica is ideal for this type of airborne and space-borne data validation and 2) the surface elevation is probably changing minimally, with respect to ice flow, snow accumulation, and surface melt, making it an ideal absolute elevation validation surface, but that there is some level of snow



redistribution (sastrugi migration) necessitating near coincident airborne surveys in space and in time to improve estimates of surface measurement precision.

## 2 Data

### 2.1 88S Traverse GPS data

We conducted a 750 km kinematic GPS survey near Amundsen-Scott South Pole Station, Antarctica using 2 tracked vehicles (PistenBullys) provided by the US Antarctic Program. The 88S Traverse departed from South Pole Station on 28 December 2017 and traveled for 4 days to the 88° S line of latitude. The traverse route then followed this line of latitude for ~300 km, before returning to South Pole Station on 10 January 2018 (Fig. 1). The kinematic GPS survey used dual-frequency Trimble NetR9 receivers recording at 1 and 2 Hz with Trimble Zephyr 2 Geodetic GNSS (TRM57971) antennas, mounted to the roof

of each PistenBully. The height of each roof-mounted GPS antenna was measured twice along the 88S Traverse; specifically, the measurement made was the distance between the antenna base plane and the bottom of the indentation of the tracks of the PistenBully into the snow. The average antenna heights for the 2 vehicles were 281.3 cm (vehicle A, 1σ standard deviation 0.9 cm) and 282.3 cm (vehicle B, 1σ standard deviation 0.4 cm). The depths of the tracks of each of the vehicles into the snow surface were measured 30 times along the traverse. The average track depths for the 2 vehicles were 6.2 cm (vehicle A, 1σ

standard deviation 1.6 cm) and 5.8 cm (vehicle B, 1σ standard deviation 1.2 cm). The antenna-height and track-depth measurements are ultimately required to calculate the distance from each of the GPS antenna phase centers to the snow surface (Fig. 2). Surveys were conducted at ∼2 m s$^{-1}$; at a 2 Hz sampling rate, this generated data points with non-uniform footprint spacing of ~1 m.

### 2.2 UAF Lidar

The University of Alaska, Fairbanks (UAF) Lidar is a laser altimeter that has typically been deployed during Alaska-based OIB campaigns (Johnson et al., 2013). The UAF Lidar surveyed the 88S Traverse on 2 separate flights (30 November and 3 December, 2017) while integrated in a commercial (Airtec) BT-67 (Basler).. The UAF system is a commercial RIEGL LMS-Q240i scanning laser altimeter transmitting in the 905 nm wavelength part of the spectrum. The system has a full scanning angle of 60°. The 2 surveys over the 88S Traverse were conducted at an aircraft speed of ∼85 m s$^{-1}$, at an altitude of ~450 m

AGL (above ground level). At this speed and altitude, and with an effective repetition rate of 10 kHz, the UAF Lidar generates a ~1.3 m diameter footprint with a total across-track swath-width of ~500 m. Within a 10 m by 10 m area, the UAF Lidar produces ~20 to 25 returns, with nearly uniform footprint spacing of ~2 m (Fig. 3).



## 2.3 Airborne Topographic Mapper (ATM)

The Airborne Topographic Mapper (ATM; Krabill et al., 2002) is a laser altimetry system used by many OIB campaigns in both the Arctic and Antarctic. ATM collected data along the 88S Traverse on 26 October 2014, while integrated on the NASA DC-8. For that deployment, ATM (version T4) consisted of a dual instrument configuration, with both wide-scan and narrow-scan lidar systems integrated simultaneously. The wide-scan lidar system is more appropriate for ice sheet surveys and has a full scanning angle of 30°. The ATM lidars are full-waveform conically-scanning system, transmitting 532 nm wavelength 6 ns pulses. Surveys were conducted at an aircraft speed of ∼100 m s$^{-1}$, at an altitude of ~450 m AGL. At this speed and altitude, and with a 3 or 5 kHz repetition rate, the wide-scan (30°) ATM lidar generates a ~1 m diameter footprint with a scanning swath width of ∼250 m. Within a 10 m by 10 m area, the wide-scan ATM produces ~6 to 8 returns, with non-uniform footprint spacing of ~5 m; data are most dense along the edge of the swath (Fig. 3).

For completeness, we note that ATM also conducted a mission that included the 88S Traverse on 26 October and 15 November 2016 (also integrated on the NASA DC-8) using the T6 version of ATM. However, analysis of these data and other flights during this campaign suggest that there is an across-track tilt within these data, that represented a 10 to 15 cm spurious elevation variation across the wide-scan ATM swath. We therefore exclude the 2016 ATM flights from further discussion.

## 3 Methods

### 3.1 88S Traverse GPS data

Following the data processing methods of Brunt et al. (2017), we post-processed 88S Traverse GPS data using PPP methods. PPP solutions use precise GPS satellite orbit and clock information to determine the kinematic GPS antenna position. Position solutions for each vehicle were determined using NovAtel's Inertial Explorer (v.8.6); processing for each vehicle was done on nearly continuous stretches of GPS data, which typically represented 1 full day of driving, or approximately 50 km. Position solutions were solved to the L1 phase center of each antenna and then referenced to the ITRF08 reference frame of the WGS84 ellipsoid. Inertial Explorer provides an estimate of a given point-position vertical accuracy; this value was used to filter suspect elevation data that had a vertical sigma of more than 8 cm.

The elevation of the snow surface, relative to the position solutions of the L1 phase center of each antenna (Fig. 2), was then determined using data from the field and the appropriate National Geodetic Survey (NGS) antenna model phase-center offset. The height of the snow surface ($h$) for each vehicle was determined based on the position solutions of the GPS antenna phase centers ($GPS_{PC}$) based on the following equation:

$$h = GPS_{PC} - h_{AntHeight} - h_{NGSmodel} + h_{TrackDepth} \,, \tag{1}$$

where $h_{AntHeight}$ is the mean distance between the antenna base plane and the indentation of the tracks in the snow (281.3 or 282.3 cm, depending on the vehicle), $h_{NGSmodel}$ is the distance between the antenna phase center and the base plane based on



the NGS model for the Trimble Zephyr 2 Geodetic antenna (4.1 cm), and $h_{TrackDepth}$ is the mean depth of the PistenBully track indentations into the snow surface (6.2 or 5.8 cm, depending on the vehicle).

## 3.2 Airborne lidar data

We obtained the UAF Lidar Scanner L1B Geolocated Surface Elevation Triplets, Version 1 data (Larsen, 2010) through the
National Snow and Ice Data Center (NSIDC) OIB Data Portal (http://nsidc.org/icebridge/portal/) for the 2 flights over the 88S Traverse area. The data files consist of latitudes, longitudes, and elevations that were derived from an integrated on-board GPS (Trimble) and inertial system (OxTS Inertial+2). GPS post-processing used PPP methods using Novatel's GrafNav (v.8.4). Processing of the lidar data, including the incorporation of the GPS and inertial data used a commercial software package (RiProcess) developed by RIEGL. These data are distributed in the ITRF08 reference frame of the WGS84 ellipsoid.

We obtained the ATM IceBridge ATM L1B Elevation and Return Strength with Waveforms, Version 1 data (Studinger, 2018) also through the NSIDC for the 2014 flight over the 88S Traverse area. The data files include latitudes, longitudes, and elevations derived from an integrated on-board GPS (Javad) and inertial system (Applanix POS AV). Differential GPS (DGPS) post-processing methods, which use a base station installed at the departure airport for this deployment. DGPS was accomplished using a software package developed by the ATM team called GITAR (GPS Inferred Trajectories for Aircraft
and Rockets; Martin, 1991). These data are distributed in the ITRF08 reference frame of the WGS84 ellipsoid.

## 3.3 Comparison strategy

We based our comparison strategy on Brunt et al. (2017). We compared the post-processed snow surface elevations from the 88S Traverse with the airborne surface elevation data, using a 'nearest-neighbor' approach. In this method, we compared the closest lidar data point to every single ground-based GPS data point. We limited our statistical analysis based on a distance
criterion, making elevation comparisons only where the lidar footprints and the GPS measurements were within a distance 1 m of one another. We then assessed the difference between the filtered GPS and ATM and UAF lidar surface elevation datasets. Once the lidar elevation data ($Lidar_{elevation}$) were associated with the GPS elevation data ($GPS_{elevation}$), the mean elevation difference ($GPS_{elevation} - Lidar_{elevation}$) is the lidar elevation bias ($B$). We note that we take the GPS elevation data to be the ground truth.

The 1σ standard deviation of this airborne lidar elevation bias ($B$) is the spread of the lidar data, or the precision, about the mean. This is also the vertical dispersion of the lidar measurements about the mean surface. The vertical dispersion, or the surface measurement precision, includes both instrument precision and geophysical properties of the surface that will affect the measurement. Instrument precision is related to factors such as instrument timing errors, geolocation knowledge, and footprint size. Geophysical properties that will affect the measurement include atmospheric effects, surface roughness, and
surface slope, although we note that our analysis is limited to a region of low (less than 1 degree) surface slope. These instrument and geophysical effects cannot be uniquely distinguished within the surface measurement precision. Ultimately,



we report elevation accuracies and surface measurement precisions as a residual, following the convention of mean bias ± 1σ standard deviation, or 0.0 ± 0.0 cm.

The lidar biases and precisions reported here are determined relative to the GPS data, which we take represent truth, with zero errors. In actuality, these errors are not zero and are a function of: 1) formal GPS errors, which include factors such as ephemeris and clock errors; 2) ionosphere and troposphere errors; 3) multipath errors; and 4) errors due to geophysical effects, such as variable snow surface strength causing variable vehicle sinking or antenna motion due to short-scale surface undulations (sastrugi).

## 4 Results

### 4.1 Ground-based GPS data evaluation

We compared the GPS position solutions of each vehicle to assess consistency of the ground-based data. After the 88S Traverse GPS data for each vehicle were post-processed, the data were then filtered based on the 8 cm vertical sigma; this reduced each GPS data set by about a third (GPS unit A: 316,948 data points were reduced to 203,603; GPS unit B: 321,689 data points were reduced to 209,253). The mean vertical sigma values for the data used in further analysis were 7.16 and 7.19 cm for ground-based GPS units A and B, respectively. We then used a nearest-neighbor approach, limited based on a 0.5 m distance criterion, and calculated the mean elevation residual between the elevation measured by the two vehicles. This residual was 1.1 ± 4.1 cm (n=26,442).

PPP GPS post-processing methods are often used in regions where long-term base-station data are not available for DGPS methods, such as the center of ice sheets. Brunt et al (2017) showed that PPP position solutions for their traverse outside of Summit Station, in the center of the Greenland Ice Sheet, were comparable to GPS position solutions using differential methods. Therefore, while we are limited with respect to the availability of permanent GPS base stations for post processing, we feel confident that our methods provide consistent and accurate results and are appropriate for this data analysis.

### 4.2 Airborne lidar evaluation

To assess the internal consistency of the UAF Lidar, we compared the processed elevation data from the 30 November 2017 flight to the 3 December 2017 flight, using a nearest-neighbor approach, limited based on a 1 m distance criteria, and calculated the mean elevation residual. This residual was 8.1 ± 10.5 cm (n>1.5 million data points). A similar assessment of internal consistency of the ATM data could not be made since our analysis was limited to a single flight, after rejecting the 2016 ATM data due to an observed across-track tilt.

### 4.3 GPS to airborne lidar results

Table 1 lists the results for the nearest-neighbor analysis of the ground-based GPS and lidar elevation comparisons for both ATM and the UAF Lidar. Both altimeters had biases less than 10 cm and surface measurement precisions less than 15 cm.



Figure 4, panel A, shows the elevations of ground-based GPS unit A. Panel B shows the difference between GPS A elevations and the 30 November UAF Lidar elevations, minus the mean difference. Panel C is similar to panel B but using the 3 December UAF Lidar data; and panel D compares the GPS data to the 2014 ATM data. Figure 5 is the same as Figure 4, but the results are relative to ground-based GPS unit B.

We examined the spatial correlation of the elevation differences calculated between the ground-based GPS data and the airborne lidar data (Motyka et al., 2010; Rolstad et al., 2009). When measurements are made within close spatial proximity of one another, they are generally similar, and measurement errors tend to be correlated; over increasing distances, measurement errors become uncorrelated. Similar to Rolstad et al. (2009), who proved a detailed summary of semivariograms, we created semivariograms of the elevation differences, which provide an assessment of the length scales where measurement errors

become independent of one another, or uncorrelated. Figures 6 and 7 provide the semivariograms for GPS unit A and B, respectively, relative to the ATM flight (top panels) and the 2 UAF Lidar flights (bottom two panels in each figure). The x axes are lag distances between the observations, in m, and the y axes are the measure of variance, in $m^2$. The red squares represent the observed elevation differences in 50 m bins and the lines represent a semivariogram model fit to these data. As the slope of the model fit to the variance asymptotes toward zero, where the lines in the figures change from blue to red, the

observations are considered to have become independent; from Figures 6 and 7, we estimate that the range at which the variance starts to be relatively unchanging, and the length scale at which measurement errors become uncorrelated, to be ~10 to 50 m. These results are based on 5 km of along-track data; semivariograms based on longer length scales (20 km) had similar results. We attribute this 10 to 50 m length scale to be associated with wind-driven surface processes and overall roughness (sastrugi), as visible in the background of Figure 3. Sastrugi causes noise about the mean surface elevation from a measurement

perspective and we assume that this is the largest source of correlated error, given the size of the footprints of the observations (1 to 2 m), the distance criteria associated with the differencing methods (1 m), and the length scale of the surface roughness associated with sastrugi (5 to 10 m).

## 5 Discussion

The 1σ mean elevation residual between the 2 GPS units for this study was 1.1 ± 4.1 cm (n=26,442). This residual compares

favorably to the GPS assessments of Brunt et al. (2017) and Kohler et al. (2013), the studies that most closely match the methods and geophysical setting presented here. Brunt et al. (2017) reported a 1σ mean elevation residual of 0.7 ± 5.7 cm, based on comparisons between 2 different passes of the traverse occurring on the same day and using the same GPS unit (n=710). Kohler et al. (2013) reported a 1σ mean elevation residual of 0.6 ± 7.5 cm, based on crossovers between 2 different GPS units during the traverses (n=1131). We attribute the quality of our GPS data to 1) the long length scale of data collection

(relative to Brunt et al., 2017) and 2) the flat surface that defined our traverse route (relative to Kohler et al., 2013). We note from Figures 4 and 5 that no discernable signal is evident in the various comparisons and we therefore attribute the differences to surface measurement noise.




While the residual between the 88S Traverse vehicles is low, it is not zero. We attribute the ~1 cm bias between our GPS datasets to uncertainties in the measurements of track depth. From Equation 1 and Figure 2, the 3 terms associated with reducing the GPS measurement to a snow-surface height are the phase center offset (which is static and common between the vehicles), the antenna height (vehicle A 281.3 ± 0.9 cm; vehicle B: 282.3 ± 0.4 cm), and the track depth (vehicle A: 6.2 ± 1.6

5    cm; vehicle B: 5.8 ± 1.2 cm). Given the uncertainties associated with the 2 field-based measurements (antenna height and track depth), we feel confident that the snow depth is the leading term in the height uncertainty.

Overall, the quality of the lidar data used in this survey was quite good. While a quantitative assessment could be made for the UAF Lidar, a similar assessment of ATM could not be made in this region, as we were limited to one flight. However, Brunt et al. (2017) analyzed ATM data from 5 different airborne campaigns, which included 5 different versions of the ATM system

(including both narrow and wide scanning data) near Summit Station, Greenland, on the relatively flat ice sheet interior, similar to this study. Their results indicated an average ATM elevation bias and surface measurement precision of -3.4 ± 6.5 cm (based on PPP post-processing, which is the method used here). Given that we are using the same lidar, with similar survey techniques, over a similar surface, we consider ATM to be a stable instrument, with data quality suitable for this application.

We note that there is a slight along-flight signature that is apparent in the UAF Lidar elevation data (Fig. 8). The signature is

visible in the southern side of the swaths of both the 30 November and 3 December 2017 datasets. Specifically, there appears to be a trough along the southern edge of the swaths that has anomalously lower elevations, relative to the surrounding edges. The magnitude is variable but based on a nearest-neighbor assessment of the overlapping region in Figure 8, where the flight line from 30 November 2017 intersected itself, the mean residual was -9.9 ± 12.7 cm. While the source of this artefact is still undetermined, it doesn't appear to be an across-track tilt. This effect on measured elevation is small (cm scale) and generally

limited to near the edge of the lidar swath (Fig. 8). These data were typically not used for ground survey GPS comparison, as the ground-based data generally intersected the center of the swath, where we believe the data quality is not compromised.

The elevation biases and surface measurement precisions of the 2 OIB lidars presented here are comparable to that of the OIB lidars assessed in Brunt et al. (2017); results based on PPP methods for both studies indicated biases that are less than ~11 cm and measurement precisions that are less than ~15 cm (Table 1 in this document and Table 2 in Brunt et al., 2017). Brunt et al.

(2017) also indicate an average ATM elevation bias and surface measurement precision of -3.4 ± 6.5 cm. From Table 1, the surface measurement precision associated with ATM over the 88S Traverse (± 14 cm; 'lower precision') was poorer quality than the average precision of ATM over the Summit Station Traverse (± 7 cm; 'higher precision') as determined by Brunt et al. (2017). These 2 assessments had a similar geophysical setting (i.e., ice-sheet interior) and similar survey strategies (GPS collection and processing methods).

We attribute the poorer surface measurement precision to the time difference between the airborne ATM campaign (October 2014) and the ground-based GPS survey (December 2017 to January 2018). Specifically, we hypothesize that these differences were associated with the transient locations of sastrugi. To assess this hypothesis, we used the same nearest-neighbor approach, described in the methods section, to compare the 2014 ATM elevation data to the 2017 UAF Lidar elevation data (Table 2). Ultimately, the difference between these 2 lidar datasets revealed a signature that was of a similar magnitude (meters) and



trend (grid SSE, or ~150º) as the sastrugi, based on observations of the sub-meter-resolution WorldView-2 satellite imagery, obtained via the Polar Geospatial Center at the University of Minnesota (Fig. 9).

Sastrugi causes noise about the mean surface elevation from a measurement perspective. Sastrugi migration between the 2014 ATM campaign and the 2017/2018 ground-based traverse would not have an impact on the surface elevation bias, as the observed differences would be averaged out and lost in surface measurement noise. The migration of the sastrugi adds components of noise on the mean surface measurement. This effect is evident in the observed larger (poorer) ATM surface measurement precision assessment.

Overall, these results suggests that the 88S Traverse route is an ideal setting to assess airborne or satellite absolute elevation accuracy (Brunt et al., 2017), as the surface was relatively unchanged between 2014 and 2018 (i.e., no distinguishable change in bias). Further, our results based on the 2014 ATM elevation dataset suggests that airborne data collected along this route are applicable to absolute elevation validation for a few years. However, results based on our comparisons between our GPS measurements and ATM suggest that when a few years has evolved between the datasets being evaluated, the surface elevation measurements become hard to reproduce; this manifests itself in a higher surface measurement precision assessment.

## 6 Conclusions

Here we present a comparison of in situ GPS elevation data and laser altimetry in preparation for ground-based and airborne validation of ICESat-2. We show that the ground-based methods for GPS data collection and processing along the 88S Traverse provide internally consistent results, with accuracies and precisions appropriate for assessing airborne lidar data and ultimately, satellite elevation data. Further, we have shown that airborne lidar data assessed here (ATM and the UAF Lidar), relative to the GPS data, show elevation biases that are comparable to results from similar instruments in similar geophysical settings. However, discrepancies between the ATM surface measurement precisions observed here, and those observed in Brunt et al. (2017) under similar ice-sheet interior conditions, suggest that the migration of sastrugi can have an adverse effect on assessments of surface measurement precision when significant time (on the order of a few years) has elapsed between surveys. Thus, absolute elevation bias can be determined with datasets from this surface that are a few seasons old, but for the best assessment of precision, comparisons need to be made with relatively coincident (spatial and temporal) datasets.

## 7 Data availability

The ground-based GPS data associated with this study are available online, as the supplement related to this article (doi:10.5194/tc-NNN-supplement). NASA ATM and the UAF Lidar data are publicly available on the NSIDC Operation IceBridge Data Portal (http://nsidc.org/icebridge/portal/). WorldView-2 imagery is available to NSF- and NASA-funded researchers via the Polar Geospatial Center at the University of Minnesota.



## Acknowledgements

We thank the NASA ICESat-2 Project Science Office for funding the field component and data analysis associated with this project. We thank the National Science Foundation, Office of Polar Programs for logistical support of the field component of this project. We thank Operation IceBridge for the data collection of the ATM and UAF Lidar datasets. We thank our deep-field mechanic and mountaineer associated with the 88S Traverse (Chad Seay and Forrest McCarthy), for ensuring that we safely completed the full Antarctic ground survey. We thank the many science support staff of the US Antarctic Program that helped make the field component of this project possible. And we thank the National Snow and Ice Data Center (NSIDC) for IceBridge data distribution. WorldView-2 imagery was provided by the Polar Geospatial Center at the University of Minnesota, which is supported by grant ANT-1043681 from the National Science Foundation.

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





**Table 1: Elevation bias and surface measurement precision (in cm), relative to ground-based GPS survey data, for ATM and UAF airborne lidar elevation data. Results are posted as *GPS_elevation − Lidar_elevation*.**

| Lidar Survey | PPP bias ± precision:<br>Relative to GPS A (cm)<br>Relative to GPS B (cm) |
|---|---|
| ATM 26 October 2014 | 2.8 ± 14.0<br>3.6 ± 14.1 |
| UAF Lidar 30 November 2017 | 0.1 ± 9.7<br>0.2 ± 9.5 |
| UAF Lidar 3 December 2017 | -9.5 ± 9.8<br>-8.0 ± 9.7 |

**Table 2: Elevation bias and surface measurement precision (in cm), between ATM and the UAF lidar. Results are posted as *ATM_elevation. − UAF_elevation*.**

| Lidar Surveys | mean bias ± 1σ standard deviation, cm |
|---|---|
| ATM 26 October 2014 vs UAF Lidar 30 November 2017 | 0.3 ± 15.0 |
| ATM 26 October 2014 vs UAF Lidar 3 December 2017 | -7.8 ± 15.1 |



**Figure 1: Map of the 88S Traverse Route, color coded based on elevation. Locations for Figures 4 – 9 are also shown. The South Pole Operational Traverse (SPoT) Route is indicated in orange. Background is the Landsat image mosaic of Antarctica (LIMA; Bindschadler et al., 2008).**





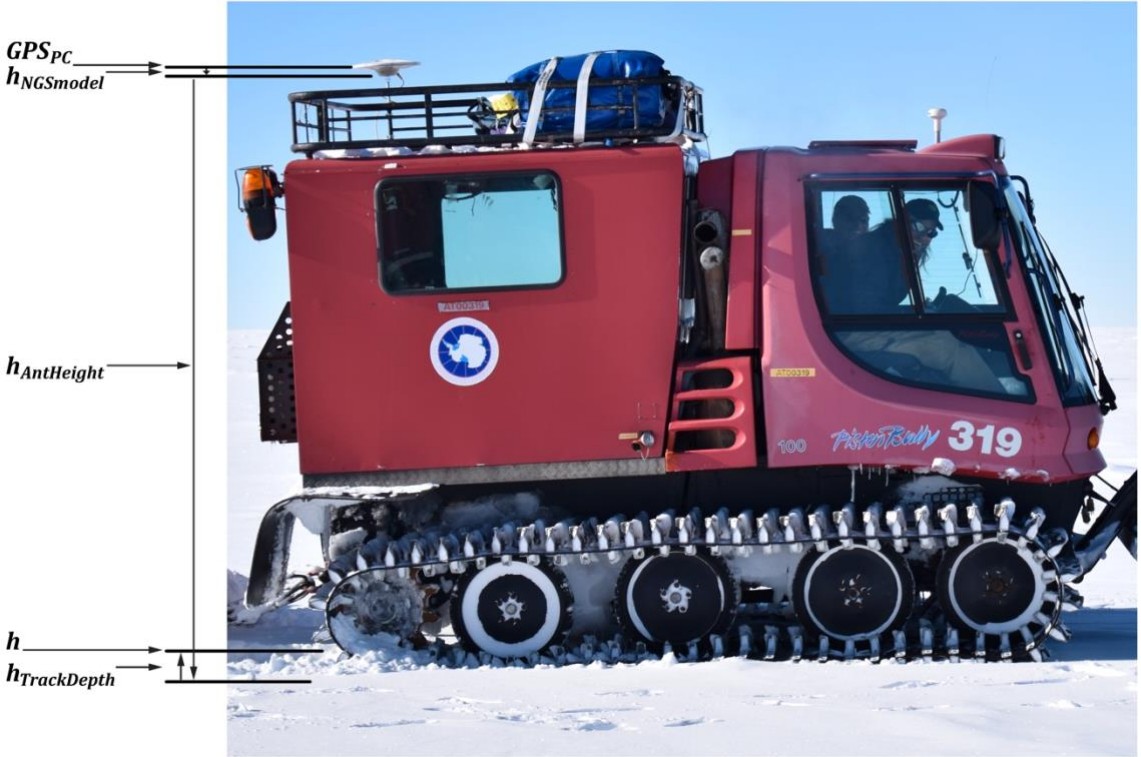

**Figure 2: The GPS antenna configuration on a PistenBully.** $GPS_{PC}$ **is the surveyed position solution to the phase center of the antenna,** $h_{NGSmodel}$ **is the NGS model distance between the antenna phase center and the antenna base plane,** $h_{AntHeight}$ **is the distance between the antenna base plane and the indentation of the tracks in the snow,** $h_{TrackDepth}$ **is the depth of the sled runners in the snow surface,**
5    **and** $h$ **is the snow surface (Eq. 1).**



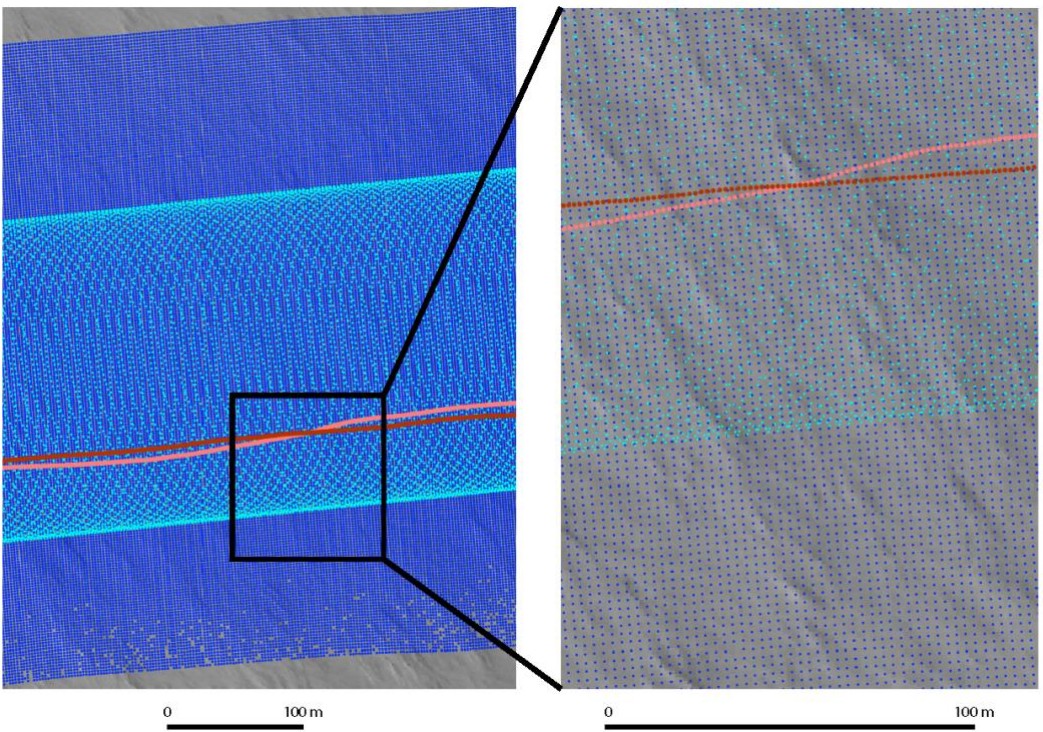

**Figure 3: Sample footprint spacing for the UAF Lidar (dark blue), ATM (cyan), and the 88S Traverse ground-based GPS data (points in shades of red). WorldView-2 imagery, copyright 2017, DigitalGlobe, Inc.**





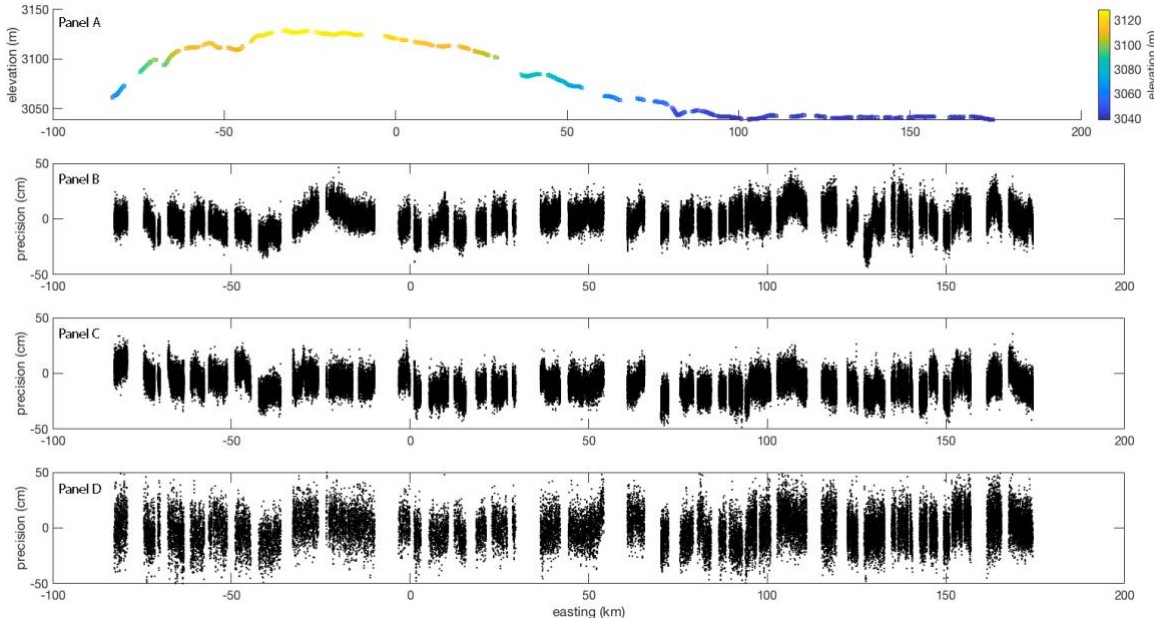

**Figure 4: Along-track elevation and elevation differences associated with GPS A. Panel A: Along-track elevation of GPS A, in m. Panel B: Elevation difference between GPS A and the UAF Lidar (30 November 2017), minus the mean difference. Panel C: Elevation difference between GPS A and the UAF Lidar (3 December 2017), minus the mean difference. And panel D: Elevation difference between GPS A and ATM (26 October 2014), minus the mean difference.**



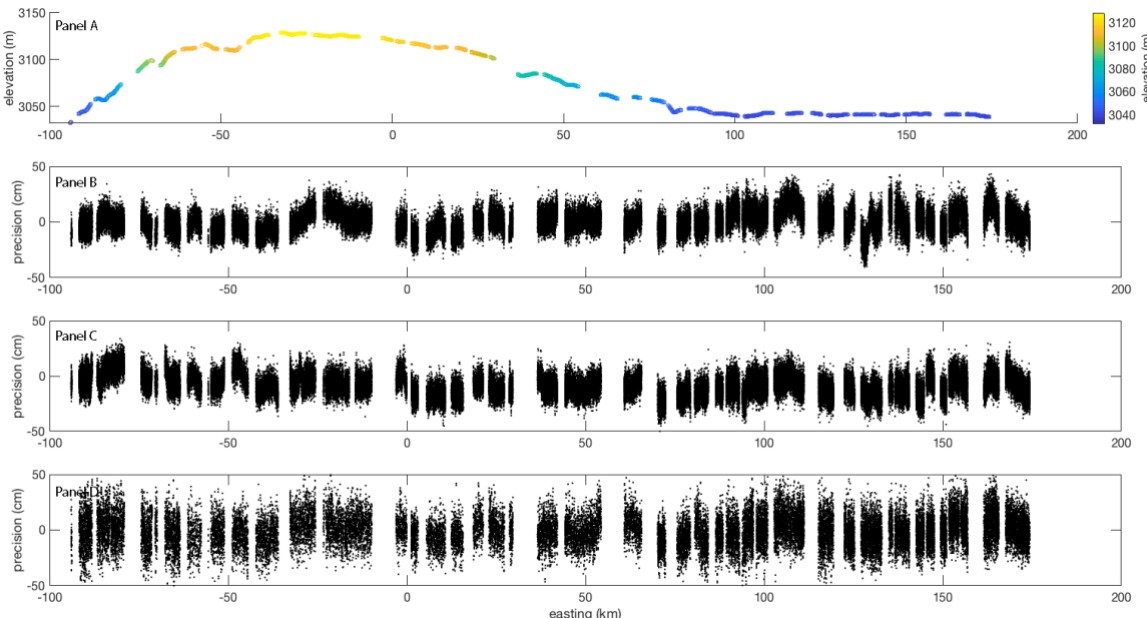

**Figure 5: Along-track elevation and elevation differences associated with GPS B. Panel A: Along-track elevation of GPS B, in m. Panel B: Elevation difference between GPS B and the UAF Lidar (30 November 2017), minus the mean difference. Panel C: Elevation difference between GPS B and the UAF Lidar (3 December 2017), minus the mean difference. And panel D: Elevation difference**
5  **between GPS B and ATM (26 October 2014), minus the mean difference.**





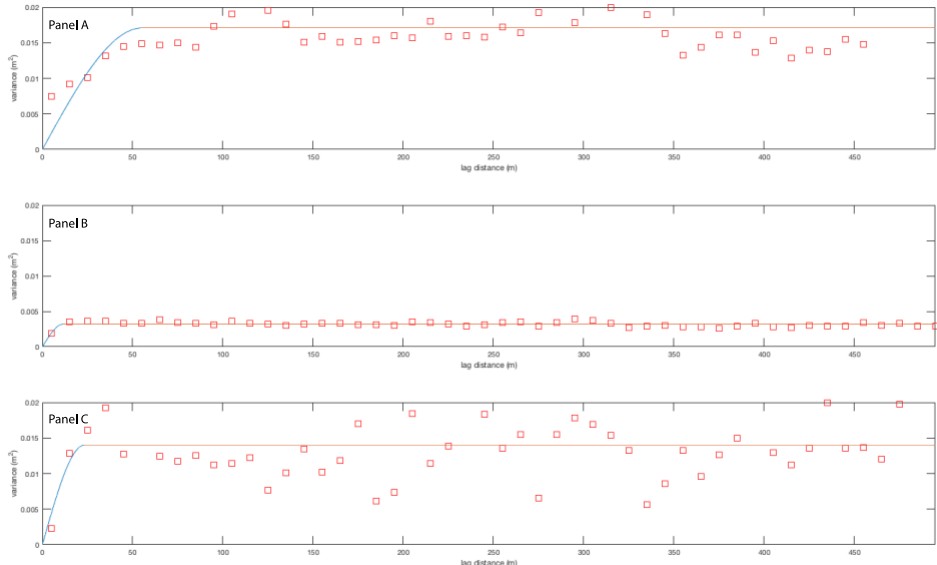

**Figure 6: Semivariograms of elevation differences between GPS unit A and elevations derived from ATM (top panel) and the UAF Lidar on 30 November 2017 (middle panel) and 3 December 2017 (bottom panel). The x axes are lag distances between the observations, in m, and the y axes are the measure of variance, in m². The red squares represent the observed elevation differences in 50 m bins and the lines represent a semivariogram model fit to these data. As the slope of the model fit to the variance asymptotes toward zero, where the lines in the figures change from blue to red, the observations are considered to have become independent.**





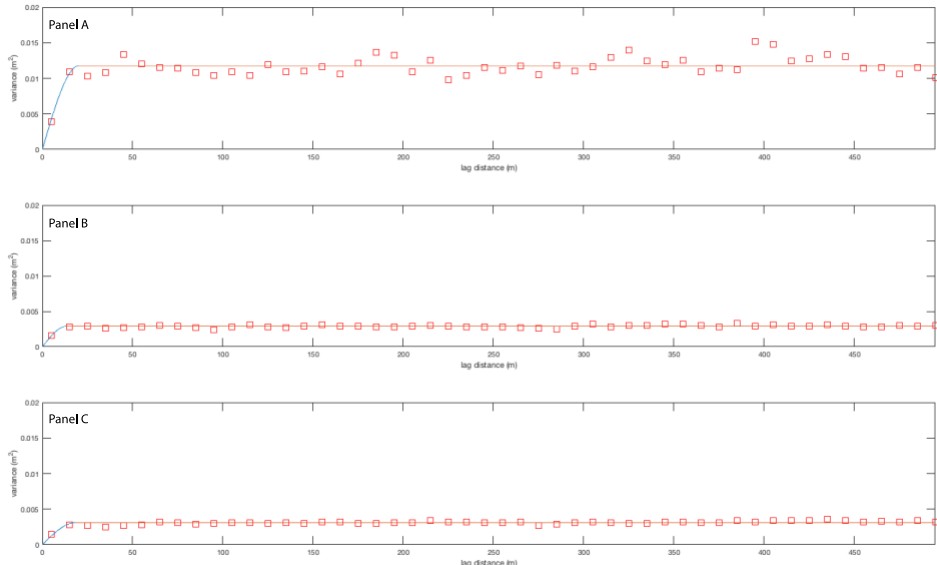

**Figure 7: Semivariograms of elevation differences between GPS unit B and elevations derived from ATM (top panel) and the UAF Lidar on 30 November 2017 (middle panel) and 3 December 2017 (bottom panel). The x axes are lag distances between the observations, in m, and the y axes are the measure of variance, in m². The red squares represent the observed elevation differences**
5   **in 50 m bins and the lines represent a semivariogram model fit to these data. As the slope of the model fit to the variance asymptotes toward zero, where the lines in the figures change from blue to red, the observations are considered to have become independent.**





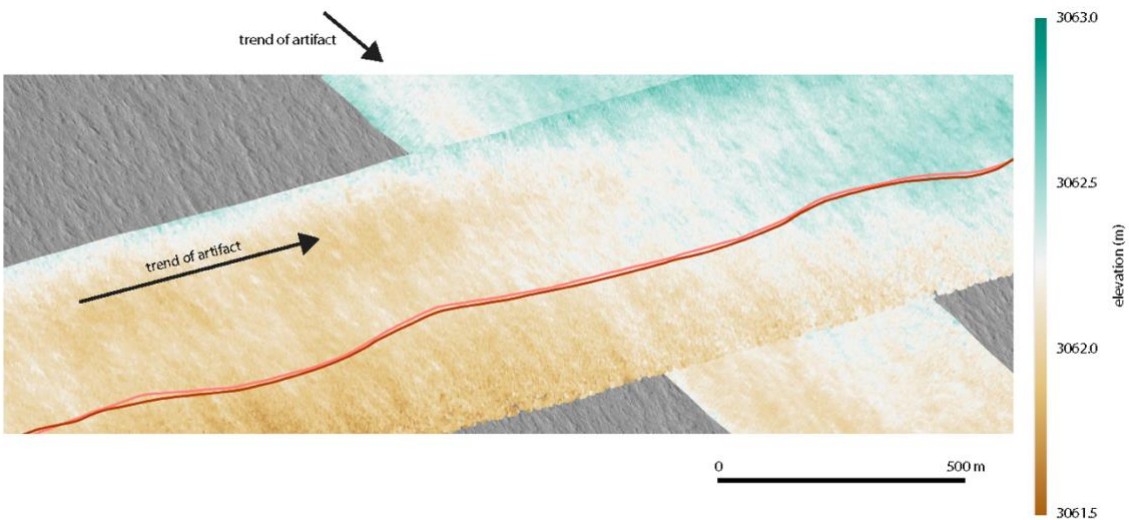

**Figure 8: Elevation data from the UAF Lidar (30 November 2017), where the flight line crossed itself. The along-track artifact in the data is visible in both passes; UAF Lidar elevations are anomalously lower within the artefact. 88S Traverse ground-based GPS data are in shades of red. WorldView-2 imagery, copyright 2017, DigitalGlobe, Inc.**





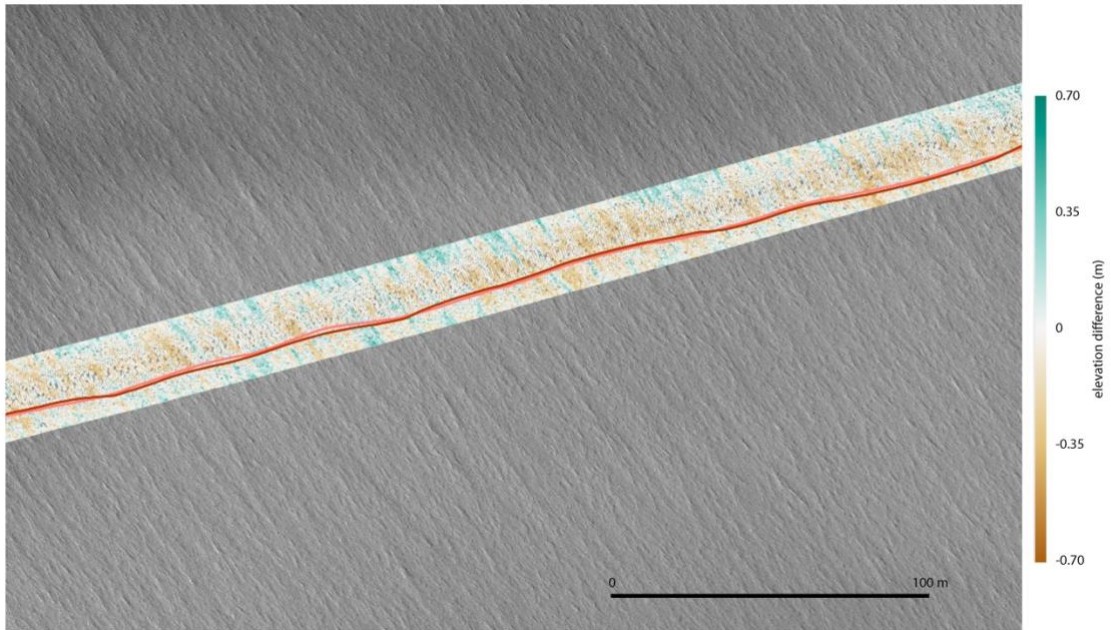

**Figure 9: Ground based GPS data (in shades of red) plotted on difference in elevation between ATM (26 October 2014) and the UAF Lidar (30 November 2017). WorldView-2 imagery, copyright 2017, DigitalGlobe, Inc.**