# Peer review of "Assessment of altimetry using ground-based GPS data from the 88S Traverse, Antarctica, in support of ICESat-2"

_The Cryosphere, 2018_

## Referee Comment (RC1) · Anonymous Referee #1 · 16 Oct 2018

The authors present new elevation data collected using kinematic GPS in Antarctica and discuss their relevance toward validation of satellite laser altimetry data from the ICESat-2 mission. The paper was concise and well-written, and I am happy with publication almost as-is.

I have three major comments:

(1) While the authors discussed elevation changes associated with sastrugi migration, there was no discussion of other surface processes, primarily firn compaction in the context of

(a) Temporary (perhaps elastic) compaction of snow/firn from the weight of the PistenBully, which might not be captured from the track depth measurements. How heavy were the PistenBullys and is this effect negligible?

(b) Climate-driven firn compaction over < seasonal to multi-year timescales showing up as elevation differences between GPS- and Operation IceBridge-derived estimates. I think it could be useful if the authors included a time series of modelled elevation change from firn processes (data available at http://www.staff.science.uu.nl/~ligte104/DATA/) at one or more locations along this transect.

(2)  The authors mentioned that there were anomalous elevations at the edge of the UAF lidar swath. This raised a red flag for me: Is there a possible connection between scan angle and elevation accuracy for the airborne lidar systems (even in campaigns where elevations did not have an across-track tilt)? The authors could include elevation differences between GPS and airborne lidar as a function of scan angle, especially since their data will likely capture a fairly wide range of scan angles. This analysis will tell us if we should only use airborne lidar data from a particular range of scan angles when comparing with ICESat-2 altimetry.

(3) It appears to me that the UAF lidar data from 3$^{rd}$ December 2017 were of lower quality that those from 30$^{th}$ November 2017 (Table 1, 2, and Figure 6 Panel C). However, biases appear to be within 1$\sigma$ uncertainties, so perhaps this difference isn't significant enough to require further discussion.

Other minor comments:

1) Page 1, Line 20: change "set to launch" to "launched". Yay!
2) Section 3.1: What cut-off angle was used in the processing?
3) It would be nice to have larger font sizes in Figures 6 and 7.

Discussion: I think a significant number of issues with using GPS data to validate airborne and satellite altimetry could potentially be mitigated in the future with the use of a terrestrial laser scanner (TLS) mounted near the GPS antenna on the PistenBullys.

---

## Referee Comment (RC2) · Anonymous Referee #2 · 5 Nov 2018

The paper reports the acquisition and processing of ground GPS data along the 88S Traverse and compares the surface elevations to those derived from airborne laser altimetry. The ultimate goal of the 88 Traverse is to validate ICESat-2 surface elevation data. Ground GPS data were processed by using Precise Point Positioning (PPP) post-processing method. The paper provides a detailed analysis of surface elevation mapping results from ground-based GPS and airborne laser altimetry.

The manuscript would benefit from including a summary on how the ground and airborne elevation datasets will be used for ICESat-2 validation. While the authors make a strong argument for using the 88S traverse because of the abundance of ICESat-2

measurements, they don't elaborate on how the data sets collected at different times and characterized by different errors will help to validate the ICESat-2.

In my opinion, the analysis presented in the paper does not provide sufficient support for the assumption that the ground GPS data depicts the real surface and thus provides ground truth for the airborne laser altimetry. While the PPP processing used for processing kinematic ground GPS data can provide vertical accuracy up to 3 cm, errors can be up to 10 cm (e.g., https://www.novatel.com/an-introduction-to-gnss/chapter-5-resolving-errors/gnss-data-post-processing/), or somewhat lower because of the filtering the authors applied. The vertical accuracy of the ATM laser altimetry points is similar to the accuracy of PPP processing. For example, Martin et. al., 2012 (Table 7) quoted a vertical accuracy of 6.6 cm with a vertical precision of 3 cm for an operating altitude of 500-750 meter above the surface.

To demonstrate the accuracy of the ground GPS surveys, the authors compared the measurements made with vehicles A and B and obtained a small elevation difference of 1.1$\pm$4.1 cm. However, this comparison does not fully characterize the absolute accuracy of surveys. Some of the errors, in particular, the ionosphere and troposphere errors, could be quite large and correlated within a short time that elapsed between data acquired with the GPS systems on the two vehicles.

Thus, the results presented in the paper do not adequately support the attribution of the elevation difference between ground GPS and airborne lidar surveys to a single source, i.e., to the random error (precision) of the airborne laser altimetry data sets. Instead, the elevation differences between the different airborne surveys (not included) as well as relative to the ground GPS survey (Figures 4 and 5) should be analyzed with the inclusion of any available calibration and/or validation information. For example, the correlated patterns in the residual elevation differences between the airborne and ground survey have characteristic spatial wavelengths of 15-50 km that corresponds to 2-7 hours with the ground speed of 2 m s$\hat{}${-2}$. This pattern might indicate modeling errors of the ionospheric and tropospheric corrections which typically decorrelate within

a few hours. Only a comprehensive analysis could shed light on the source of the errors and attribute them to the different surveys.

Detailed comments:

General comments:

- The description of the horizontal and vertical reference frames is somewhat confusing in the paper. The reference of the geographic coordinates is not mentioned and the elevations are described to be referenced to the ITRF reference frame of the WGS-84 ellipsoid. The correct information for ATM laser altimetry is that elevations are given in the ITRF08 reference frame, and geographic coordinates are referenced to the WGS84 ellipsoid (e.g, https://nsidc.org/data/ILATM1B). Similar information should be included for the other data sets.

Page 1, lines 19-25: of course, these sentences should be in past tense after the successful launch of ICESat-2

Line 24: include the spatial scale of the ICESat-2 mission requirement of 0.4 cm/yr

Page 2, lines 13-18: according to the authors the ground tracks will be spaced randomly in time (line 15) and evenly (line 16). It is a confusing description and needs clarification. Also, how are the ground tracks distributed relative to the ICESat-2 sub-cycles?

Lines 24-30: these statements are too concise to understand without consulting the referenced studies. For example, what are the surveys to which the surface measurement precisions refer?

Lines 32-33: use the words spaceborne and airborne consistently, preferably without a hyphen

Line 33: does the surface elevation change minimally or not? In my opinion, enough is known about the elevation changes of the East Antarctic Plateau to make a definitive

statement and even quantify the changes

Page 3, section 2.1: more details about the kinematic GPS surveys should be included. What was the typical distance (space and/or time) between the two vehicles during data acquisition? Did they always travel in the same order or did they switch place? Was there any static GPS collected?

Line 10-12: How many times was the antenna height measured? If only twice (line 10), the standard deviation mentioned in line 12 is not meaningful.

UAF lidar: the configuration of the UAF lidar – line scanner – should be mentioned.

ATM: please refer to the information included in Martin et al., (2012) about the accuracy and precision of the ATM system, instead of the somewhat outdated Krabill et al., 2002

Page 4, lines 11-14: What is the difference between the ATM T4 and T6 systems? What was the flying height of those ATM flights that are not included? Is there a study/personal communication that can be referenced about the presence and cause of the across-track tilt, i.e., uncorrected attitude error?

Page 5, lines 4-15: it would be useful to repeat the data acquisition times in this section.

Line 11: use "geographic coordinates" instead of latitudes and longitudes.

Line 26: what is the spatial scale to compute the standard deviation? Why using 1-sigma instead of the more widely used 2-sigma or 3-sigma?

Page 6, line 24: which flight provided higher elevations?

Page 7, lines 5-22: Were the biases removed from the differences before computing the variograms? Was there any additional preprocessing applied? Using the standard terminology for the description of the variograms (sill, range, etc.) would improve the section. The difference between elevations from GPS unit A and ATM (Figure 6, top panel) has a non-zero nugget effect, while the other differences do not exhibit any. What could cause this difference?

Lines 30-32: contrary to the statement about "no discernable signal is evident," in my opinion, all differences show a clear and correlated spatial variation, which requires an explanation.

Figures:

Figure 1: adding the easting/northing would allow cross-referencing this figure with figures 4-5.

Figure 3: explain the significance and meaning of the different shades of red

Figure 4-5: X and Y axis titles should use larger fonts. The vertical axis should refer to the computed quantity, eg., "Residual elevation difference (m)" rather than the interpretation "Precision." Adding the data sources, e.g., GPS A minus UAF Lidar Dec 3, 2017, as a title to each panel would make it possible to view the figure without reference to the figure caption

Figure 6-7: axis titles and labels are tiny, should be larger. The figure caption is too descriptive, use the standard terminology of variograms (sill, range, nugget) instead. Figure 8: describe the artifact in the title, e.g., narrow ridge of elevations parallel with flight direction

References:

Martin, C. F., Krabill, W. B., Manizade, S. S. & Russell, R. L. Airborne Topographic Mapper Calibration Procedures and Accuracy Assessment. NASA Technical Memorandum 2012-215891, (2012).

---

## Author Comment (AC1) · 3 Dec 2018

**Response to Comments from Reviewer 1**

AUTHORS: *We thank the Reviewer for their comments; edits based on their input have improved this manuscript.*

The authors present new elevation data collected using kinematic GPS in Antarctica and discuss their relevance toward validation of satellite laser altimetry data from the ICESat-2 mission. The paper was concise and well-written, and I am happy with publication almost as-is.
I have three major comments:

(1) While the authors discussed elevation changes associated with sastrugi migration, there was no discussion of other surface processes, primarily firn compaction in the context of

(a) Temporary (perhaps elastic) compaction of snow/firn from the weight of the PistenBully, which might not be captured from the track depth measurements. How heavy were the PistenBullys and is this effect negligible?
AUTHORS: The reviewer is correct: Our track-depth measurements almost certainly would not capture the elastic effect you are describing. PistenBullys weigh about 10,000 kg, spread over a 10 m length and 2 m width scale. While the elastic effect of that weight pushing temporarily on the firn might not be negligible, our suspicion is that the impact is below our leading error term, which is the track depth uncertainty and non-uniformity (~6 cm ± 1.5 cm). We also note that if the elastic effect is less than the precision of our GPS data (~4 cm) then we may not be able to observe this without creative survey methods. If extant, such an elastic effect would present as a negative bias of the vehicles relative to a non-invasive data set, such as airborne or spaceborne lidar. We intend to experiment with some creative survey methods, as time allows, at the start of this field season. But currently, this is beyond the scope of this paper. Further, we have other strategic plans to reduce the track-depth uncertainty; this includes running the GPS from the sled (which seemed to float at a consistent depth in the snow), as opposed to the PistenBullys (where the tracks seemed to dig into the snow at varying depths).

(b) Climate-driven firn compaction over < seasonal to multi-year timescales showing up as elevation differences between GPS- and Operation IceBridge-derived estimates. I think it could be useful if the authors included a time series of modelled elevation change from firn processes (data available at http://www.staff.science.uu.nl/~ligte104/DATA/) at one or more locations along this transect.
AUTHORS: The reviewer is correct that time variable climate-driven firn compaction could manifest as an elevation difference between our measurements and the airborne data presented here. The seasonal effect is less of a concern as the elevation response to climate-driven firn processes are effectively annual, and in this region should be driven by the annual cycle of temperature. As the UAF data were nearly coincident in time with the GPS data collection, the impact of seasonal firn processes on elevation should be minimal. The ATM data was collected with a larger offset with respect to time of year (~2 months) which may be impacting our comparisons. The secular trend in elevation owning to climate-driven firn processes is a thornier problem, as in this sector the climate drivers (accumulation rate, temperature) are poorly known. Both the secular and annual trends in elevation owing to firn processes will be a primary concern in our second paper on the topic, which will compare two seasons of ground-based data collection with airborne and spaceborne lidar data. We have added the following text to the document:
*"We note that our analysis does not attempt to account for elevation changes due to the temperature- and accumulation-rate-driven effects of firn compaction (Li and Zwally, 2015). In this region, we expect variation in firn compaction rate to be driven by changes in firn temperature, which have a large seasonal amplitude and a much smaller secular trend. As the firn warms each austral spring, the surface elevation along our traverse should decrease. Since the UAF lidar data and ground-based GPS data were collected within a month, we expect firn compaction to have a negligible effect on our results. Conversely, the ~2 month seasonal lag between the*

*ATM and GPS data collection means that we may be sensitive to the seasonality of firn compaction rate, as well as any secular trend over the 4 year interval between these data sets."*

(2) The authors mentioned that there were anomalous elevations at the edge of the UAF lidar swath. This raised a red flag for me: Is there a possible connection between scan angle and elevation accuracy for the airborne lidar systems (even in campaigns where elevations did not have an across-track tilt)? The authors could include elevation differences between GPS and airborne lidar as a function of scan angle, especially since their data will likely capture a fairly wide range of scan angles. This analysis will tell us if we should only use airborne lidar data from a particular range of scan angles when comparing with ICESat-2 altimetry.
AUTHORS: We do not think that scan angle is the root cause of this artifact. We note that the trough is isolated to one side (in our case the southern side) of the transect. Our expectation is that if the artifact was associated with scan angle, some matching form of the trough would also appear on the other (northern edge).

(3) It appears to me that the UAF lidar data from 3rd December 2017 were of lower quality that those from 30th November 2017 (Table 1, 2, and Figure 6 Panel C). However, biases appear to be within $1\sigma$ uncertainties, so perhaps this difference isn't significant enough to require further discussion.
AUTHORS: Our experience is that mature airborne lidar accuracies and precisions are generally under 10 cm ± 15 cm. Therefore, because data from both flights fell within this general rule of thumb, we put no stock in one having a lower bias; we do not think that this suggests that that flight is significantly better than the other. The reviewer makes a great point; and there is language (some new language) that touches on this in the text (e.g., first few lines of section 4.3, which we have augmented):
*"Both altimeters had elevation biases less than 10 cm and surface measurement precisions less than 15 cm; we note that these values are similar to those in Brunt et al. (2017), which is a similar study in a similar geophysical setting."*

Other minor comments:

1) Page 1, Line 20: change "set to launch" to "launched". Yay!
AUTHORS: Done! Yippee!!

2) Section 3.1: What cut-off angle was used in the processing?
AUTHORS: Good addition. We added the following text:
*"We used a GPS satellite elevation mask, or a cut-off angle, of 7.5 degrees, to minimize the effects of multipath."*

3) It would be nice to have larger font sizes in Figures 6 and 7.
AUTHORS: This was done in conjunction with other edits suggested by Reviewer 2.

Discussion: I think a significant number of issues with using GPS data to validate airborne and satellite altimetry could potentially be mitigated in the future with the use of a terrestrial laser scanner (TLS) mounted near the GPS antenna on the PistenBullys.
AUTHORS: We plan on attempting to make a change in the coming season that starts to get at the reviewer's point. We will try to integrate a downward looking laser, next to the GPS antenna, off of the side of the rear of the sled. This should give us an 'along-track' profile of surface roughness. While this is not provide the 3D sense of roughness around our GPS survey data (that the TLS would provide), it may help beat down the error term associated with track depth.

Thank you again,
Brunt, Neumann, Larsen

---

## Author Comment (AC2) · 3 Dec 2018

**Response to Comments from Reviewer 2**

AUTHORS: *We thank the Reviewer for their attention to detail; their input has contributed significantly to this manuscript.*

The paper reports the acquisition and processing of ground GPS data along the 88S Traverse and compares the surface elevations to those derived from airborne laser altimetry. The ultimate goal of the 88 Traverse is to validate ICESat-2 surface elevation data. Ground GPS data were processed by using Precise Point Positioning (PPP) post-processing method. The paper provides a detailed analysis of surface elevation mapping results from ground-based GPS and airborne laser altimetry.

The manuscript would benefit from including a summary on how the ground and airborne elevation datasets will be used for ICESat-2 validation. While the authors make a strong argument for using the 88S traverse because of the abundance of ICESat-2 measurements, they don't elaborate on how the data sets collected at different times and characterized by different errors will help to validate the ICESat-2.

AUTHORS: We've added a long paragraph to the end of the discussion that ties together the ground-based and airborne datasets and their relation to ICESat-2 validation. We mention the strength of this dataset and that the methods presented here provide a framework for data analysis in the future:

*"Data collected from the 88S Traverse (and data collected on subsequent surveys of the same route) will provide 300 km of in situ data for direct comparison with ICESat-2 elevation data products. The GPS data collection strategies and post-processing methods presented here provide accurate and precise data for such an assessment. Further, the data analysis presented here provides guidance on how to make similar comparisons between ground-based and satellite elevations, given the satellite footprint size and associated rejection criteria. Approximately 3 to 4 ICESat-2 reference ground tracks will intersect this region daily to produce many statistical crossover points between the GPS and ICESat-2 datasets. While the crossover points represent only a small segment of along-track ICESat-2 data, the analysis will be based on data from many ICESat-2 reference ground tracks over the course of the entire satellite mission. Thus, the analysis of the derived ICESat-2 bias and surface measurement precision relative to these GPS data will provide an assessment of how ICESat-2 is performing through time, independent of errors associated with single orbits or single points in time. Results of Brunt et al. (2017) and results presented here also provide an assessment of the accuracy and surface measurement precision of 3 airborne lidars that NASA has routinely deployed over the ice sheets (ATM, LVIS, and the UAF lidar). With a statistical understanding of how these instruments perform on the relatively flat ice-sheet interiors, longer flight lines can be constructed over similar such flat ice-sheet surfaces to create better statistics associated with comparisons using long length scales of along-track ICESat-2 data. In summary, the strategic location of the ground-based 88S Traverse provides a validation of ICESat-2 that is independent of the errors that are correlated with respect to most satellite time scales, and these ground-based data provide s with a better understanding of airborne lidars that will survey longer length scales of data, for better satellite error statistics."*

In my opinion, the analysis presented in the paper does not provide sufficient support for the assumption that the ground GPS data depicts the real surface and thus provides ground truth for the airborne laser altimetry. While the PPP processing used for processing kinematic ground GPS data can provide vertical accuracy up to 3 cm, errors can be up to 10 cm (e.g., https://www.novatel.com/an-introduction-to-gnss/chapter-5- resolving-errors/gnss-data-post-processing/), or somewhat lower because of the filtering the authors applied. The vertical accuracy of the ATM laser altimetry points is similar to the accuracy of PPP processing. For example, Martin et. al., 2012 (Table 7) quoted a vertical accuracy of 6.6 cm with a vertical precision of 3 cm for an operating altitude of 500-750 meter above the surface.

To demonstrate the accuracy of the ground GPS surveys, the authors compared the measurements made with vehicles A and B and obtained a small elevation difference of 1.14.1 cm. However, this comparison does not fully characterize the absolute accuracy of surveys. Some of the errors, in particular, the ionosphere and troposphere errors, could be quite large and correlated within a short time that elapsed between data acquired with the GPS systems on the two vehicles.

AUTHORS: These 2 comments are very similar and are addressed in this single response. The reviewer makes valid points about our results and the limitations of PPP post-processing. We note that we are limited with respect to options for processing to the best possible position solution; there are very long baselines to any ground-based permanent station, which removes the differential GPS (DGPS) option. Therefore, we tried to make an assessment of the quality of these data, with only the 2 GPS data streams to work with. The crossover comparison seemed like the best method for making this assessment, and follows similar approaches of Siegfried et al. (2011), Kohler et al. (2013) and Brunt et al. (2017).

We specifically state that we take our GPS measurements to "*represent truth, with zero errors. In actuality, these errors are not zero and are a function of: … 2) ionosphere and troposphere errors; ….*" Thus, we feel that we made an effort to highlight the issue that the reviewer mentions.

But we fully agree with the reviewer's assessment that the GPS A to GPS B statistics are probably better than would be if the vehicles surveyed the region on different days (which is not possible for operational safety reasons). Therefore, we put further caveats on the 'truth' statement above (in Section 3) and in the Discussion section, where we discuss the results of the GPS data analysis (Section 5):

Section 3:

"*represent truth, with zero errors. In actuality, these errors are not zero and are a function of: … 2) ionosphere and troposphere errors; … We note that given the short distance between the 2 survey vehicles, our results are somewhat blind to the full magnitude of the error terms that can be correlated on short time scales, such as those associated with the ionosphere and troposphere.*"

Section 5:

"*As stated above, we note that we are blind to errors introduced by the close spatial coincidence of the GPS receivers (~50 m) and to those introduced by the common processing of the GPS data. Errors in ionospheric or tropospheric modeling would impact both GPS-based data sets similarly and would introduce a bias between the GPS measurements and the actual ice sheet surface.*"

In light of this, and subsequent conversations, we plan on attempting to make a change in the coming season that starts to get at the reviewer's point. We will try to survey the first 10 or so km of the traverse route a few days prior to the actual start of the full traverse. These data will help assess the magnitude of the troposphere errors.

We note that we also added the numbers of Martin et al. (2012) in the Discussion section.

Thus, the results presented in the paper do not adequately support the attribution of the elevation difference between ground GPS and airborne lidar surveys to a single source, i.e., to the random error (precision) of the airborne laser altimetry data sets. Instead, the elevation differences between the different airborne surveys (not included) as well as relative to the ground GPS survey (Figures 4 and 5) should be analyzed with the inclusion of any available calibration and/or validation information. For example, the correlated patterns in the residual elevation differences between the airborne and ground survey have characteristic spatial wavelengths of 15-50 km that corresponds to 2-7 hours with the ground speed of 2 m s$^{-2}$. This pattern might indicate modeling errors of the ionospheric and tropospheric corrections which typically decorrelate within a few hours. Only a comprehensive analysis could shed light on the source of the errors and attribute them to the different surveys.

AUTHORS: Using these 3 datasets, we do not feel that it is not possible to definitively prove that any particular data set measures the true surface elevation, owing in part to some of the reviewer's concerns (e.g., tropospheric effects on the ground-based dataset). However, by assuming that one data set represents the surface elevation (i.e., 'the truth') we can evaluate the corresponding data sets in reference to it. See comments above on how we have softened language associated with taking our ground-based data as the truth and then making assessments of the other datasets based on that.

Detailed comments:

General comments:

- The description of the horizontal and vertical reference frames is somewhat confusing in the paper. The reference of the geographic coordinates is not mentioned and the elevations are described to be referenced to the ITRF reference frame of the WGS-84 ellipsoid. The correct information for ATM laser altimetry is that elevations are given in the ITRF08 reference frame, and geographic coordinates are referenced to the WGS84 ellipsoid (e.g, https://nsidc.org/data/ILATM1B). Similar information should be included for the other data sets.
AUTHORS: We've made these clarifications for all 3 references to ITRF (in the traverse data, and both lidar datasets).

Page 1, lines 19-25: of course, these sentences should be in past tense after the successful launch of ICESat-2
AUTHORS: The other reviewer caught this too! We've made this change (although if we had used past tense in the submitted version of this paper, we surely would have jinxed ourselves!)

Line 24: include the spatial scale of the ICESat-2 mission requirement of 0.4 cm/yr
AUTHORS: While most ICESat-2 requirements have along-track length scales, this particular requirement is a 'whole ice sheet' requirement. But we did add text associated with the time requirement (on an annual basis).

Page 2, lines 13-18: according to the authors the ground tracks will be spaced randomly in time (line 15) and evenly (line 16). It is a confusing description and needs clarification. Also, how are the ground tracks distributed relative to the ICESat-2 subcycles?
AUTHORS: Great point. We think that this edit mitigates your primary concern:
*"Therefore, the 20% of the 1387 unique tracks that are intersected by this survey represent data collected from the whole 91-day orbital cycle, and not a specific 20% of the cycle. Further, since the ground tracks intersected by the 88S Traverse are spread throughout the 91-day orbital cycle, …"*

ICESat-2 does have a ~30-day subcycle to meet the sea ice requirement of a monthly sea ice freeboard data product. The requirement specifically stipulates 35 km spacing at 70° latitude. But there is no operational impact of this subcycle requirement and it's not germane to the ice sheet analysis described here. We have therefore decided not to introduce this concept, as it may confuse readers.

Lines 24-30: these statements are too concise to understand without consulting the referenced studies. For example, what are the surveys to which the surface measurement precisions refer?
AUTHORS: We have broken this section into 2 distinct pieces (Brunt et al., 2018, and Kohler et al., 2013) and added text for clarity; the text addresses numbers of campaigns (for Brunt et al.) or length-scales of data (i.e., 'the entire traverse', for Kohler et al.):
*Using Precise Point Positioning (PPP) post-processing methods for 6 ground-based GPS surveys, elevation biases for the associated 6 ATM airborne surveys (conducted between 2009 and 2016) ranged from -10.8 to 0.8 cm, while surface measurement precisions were equal to or better than 8.7 cm. Using the same methods for 2*

*ground-based GPS surveys, elevation biases for 2 LVIS airborne surveys (conducted in 2007 and 2010) ranged from -2.7 to 8.2 cm, while surface measurement precisions were equal to or better than 6.1 cm. Their results suggest that for a flat, relatively smooth and homogeneous surface, these altimeters provide consistent results, which are required for an airborne component of an ICESat-2 validation strategy. Kohler et al. (2013) collected 5000 km of ground-based kinematic GPS data along the Norway-USA East Antarctic Traverse, in 2 different vehicles and over the course of 2 different field campaigns, for direct comparison with ICESat elevation data from all of the satellite campaigns typically used for data analysis (e.g., L2A through L2E). Using PPP post-processing methods, elevation biases for ICESat, based on ground-based data from the entire traverse, ranged from -12 to -2 cm, while surface measurement precisions were equal to or better than 15.8 cm. Their results were based on cross-over analysis between ground-based measurements and the last 2 years of ICESat data."*

Lines 32-33: use the words spaceborne and airborne consistently, preferably without a Hyphen
AUTHORS: Good edit. We went with 'airborne' and 'spaceborne' throughout.

Line 33: does the surface elevation change minimally or not? In my opinion, enough is known about the elevation changes of the East Antarctic Plateau to make a definitive statement and even quantify the changes
AUTHORS: The authors would argue that very little is known about this region with respect to overall change and specifically to the terms listed (ice flow, snow accumulation, and surface melt). This area is generally within the satellite pole hole of most polar-orbiting satellites, and surface measurements are sparse in the region. Thus, we are comfortable with the language used here.

Page 3, section 2.1: more details about the kinematic GPS surveys should be included. What was the typical distance (space and/or time) between the two vehicles during data acquisition? Did they always travel in the same order or did they switch place? Was there any static GPS collected?
AUTHORS: All great additions. We have added text throughout section 2.1 that addresses all of these comments. Plus, the other reviewer asked about cut-off angles in processing. We have added these details.

Line 10-12: How many times was the antenna height measured? If only twice (line 10), the standard deviation mentioned in line 12 is not meaningful.
AUTHORS: Good point. We removed the standard deviations and instead listed the measurements, e.g.: *"The average antenna heights for the 2 vehicles were 281.3 cm (vehicle A, 280.7 cm and 281.9 cm) and …"*

UAF lidar: the configuration of the UAF lidar – line scanner – should be mentioned.
AUTHORS: Good point. We added this text.

ATM: please refer to the information included in Martin et al., (2012) about the accuracy and precision of the ATM system, instead of the somewhat outdated Krabill et al., 2002
AUTHORS: We have added the Martin reference in this section of the paper; and the details of their vertical accuracy/precision results (6.6 ± 3 cm) was added to page 8, where it was more applicable.

Page 4, lines 11-14: What is the difference between the ATM T4 and T6 systems? What was the flying height of those ATM flights that are not included? Is there a study/personal communication that can be referenced about the presence and cause of the across-track tilt, i.e., uncorrected attitude error?
AUTHORS: ATM: Our group is not knowledgeable on the differences between the T4 and T6 versions of ATM, and given that we do not move forward with the 2016 data, this is a bit outside the scope of the paper. We have added the survey altitude (~450 m AGL) and a personal communication remark (Michael Studinger) associated with cross-track tilt; but as of writing, it is our understanding that the source was still unknown.

Page 5, lines 4-15: it would be useful to repeat the data acquisition times in this section.
AUTHORS: This is a good addition. We were able to add times for ATM, but the UAF lidar do not have associated times; but we added text that could point future users to the exact files.

Line 11: use "geographic coordinates" instead of latitudes and longitudes.
AUTHORS: Done.

Line 26: what is the spatial scale to compute the standard deviation? Why using 1- sigma instead of the more widely used 2-sigma or 3-sigma?
AUTHORS: All of the biases and standard deviations presented here are calculated using the full run of the dataset (~300 km). With respect to the use of 1-sigma, we are building off of results from other ICESat and ICESat-2 validation efforts (e.g., Fricker et al., 2005, Kohler et al., 2013, Borsa et al., 2014, and Brunt et al., 2016). Results presented here can therefore be directly compared with the results from the other papers. We note that all of these papers are very specific about exactly what they are presenting (e.g., but stating 'SD', or '1-sigma').

Page 6, line 24: which flight provided higher elevations?
AUTHORS: Data from the 30 November 2017 flight were lower than data from the 3 December 2017 flight; we've added this text to the manuscript.

Page 7, lines 5-22: Were the biases removed from the differences before computing the variograms? Was there any additional preprocessing applied? Using the standard terminology for the description of the variograms (sill, range, etc.) would improve the section. The difference between elevations from GPS unit A and ATM (Figure 6, top panel) has a non-zero nugget effect, while the other differences do not exhibit any. What could cause this difference?
AUTHORS: The biases were not removed prior to the analysis, and no further pre-processing was applied. We have added the 'sill' and 'range' terms to a few places in the text.
With respect to GPS A and ATM (which is now the bottom panel of Fig 6 to be more uniform with other figures and the layout of the text), the reviewer makes a good point. We note that this is also the comparison with the greatest range, at ~50 m; the others tend to be <20 m. We assume that this is associated with the sastrugi migration, since it has to be either (or a combination of) measurement error or small scale variation (smaller than sampling scale). The sastrugi migration would be in the approximate size/distance scale. But this is a bit of speculation. The effect of sastrugi on the differences in the overall results is discussed more coherently in the 'Discussion' section.

Lines 30-32: contrary to the statement about "no discernable signal is evident," in my opinion, all differences show a clear and correlated spatial variation, which requires an explanation.
AUTHORS: We respectfully disagree. We see three instances where there is a spatial correlation across panels 5b, 5c, and 5d (at 0 easting, 110 easting and 170 easting). These could be the result of many things, but track depth changes would be an obvious candidate. And we have already commented that we feel that this is the leading error term in our survey and analysis (Section 5).

We note that this text was ultimately removed, as it was not relevant to the GPS-to-GPS paragraph topic.

Figures:

Figure 1: adding the easting/northing would allow cross-referencing this figure with figures 4-5.
AUTHORS: Good addition. This has been added.

Figure 3: explain the significance and meaning of the different shades of red
AUTHORS: The red distinguishes the 2 GPS units. We have added text for clarification:
… *"and the 88S Traverse ground-based GPS data are in shades of red (GPS A is in light red, while GPS B is in dark red)."*

Figure 4-5: X and Y axis titles should use larger fonts. The vertical axis should refer to the computed quantity, eg., "Residual elevation difference (m)" rather than the interpretation "Precision." Adding the data sources, e.g., GPS A minus UAF Lidar Dec 3, 2017, as a title to each panel would make it possible to view the figure without reference to the figure caption
AUTHORS: All great changes; these have been made.

Figure 6-7: axis titles and labels are tiny, should be larger. The figure caption is too descriptive, use the standard terminology of variograms (sill, range, nugget) instead.
AUTHORS: We have made these changes to the figure. We removed some of the figure-caption text and added sill, range, and nugget to the main text.

Figure 8: describe the artifact in the title, e.g., narrow ridge of elevations parallel with flight direction
AUTHORS: We added language to the figure caption: "…*UAF Lidar elevations are anomalously lower within the artefact and manifest as a narrow trough, parallel to the direction of flight*."

References:

Martin, C. F., Krabill, W. B., Manizade, S. S. & Russell, R. L. Airborne Topographic Mapper Calibration Procedures and Accuracy Assessment. NASA Technical Memorandum 2012-215891, (2012).
AUTHORS: We have added this reference.

Thank you again,
Brunt, Neumann, Larsen